# Predicting the spatial pattern of land use change and carbon storage in Xinjiang: A Markov-FLUS-InVEST model approach

**Mengting Jin[1], Xingxing Duan[1], Yunfei Zhang[1]\*, Quan Xu[1,2]\***

**1** China Geological Survey Urumqi Comprehensive Survey Center on Natural Resources, Urumqi, China,
**2** College of Ecology and Environment, Xinjiang University, Urumqi, China

\* zhangyunfei@mail.cgs.gov.cn (YZ); xuquan@mail.cgs.gov.cn (QX)

## Abstract

Land-use changes significantly influence carbon storage capacity by altering the structure, layout, and function of terrestrial ecosystems. Predicting the relationship between future land-use changes and carbon storage is essential for optimizing land-use patterns and making rational, ecology-based decisions. Using multi-period land-use data from Xinjiang, we analyzed the spatial pattern of carbon storage. Based on land-use change patterns in Xinjiang from 2000 to 2020, we coupled the Markov-Future Land Use Simulation (FLUS)-Integrated Valuation of Ecosystem Services and Tradeoffs (InVEST) model to simulate and predict land-use spatial patterns in Xinjiang for 2035 under two scenarios: natural growth and ecological protection. Carbon storage and its spatiotemporal dynamic changes under these scenarios were evaluated, and the Geodetector was employed to analyze the spatial heterogeneity of carbon storage from a statistical perspective, revealing the influence of various driving factors. The results showed that: (1) From 2000–2020, grassland and unused land were the primary land-use types in Xinjiang, accounting for over 28.85% and 60.17% of the total area, respectively. By 2035, cropland, forest, water, and construction land areas are expected to increase, while grassland and unused land areas are projected to decrease. Under the ecological protection scenario, cropland, forest land, and grassland—major main contributors to carbon storage—will be effectively conserved to some extent. (2) From 2000 to 2020, Xinjiang's carbon storage capacity exhibited an overall increasing trend, with a cumulative increase of $137.515 \times 10^5$ t and a growth rate of 1.58%. However, this capacity is projected to decline by 2035, with an estimated reduction of $168.344 \times 10^5$ t compared to that in 2020. Ecological protection is anticipated to mitigate this decline, increasing carbon storage by $13.227 \times 10^5$ t relative to the natural growth scenario. (3) Geodetector analysis indicated that land-use types had the greatest carbon storage explanatory power for carbon storage (q = 0.80), followed by soil types (q = 0.41), net primary productivity (q = 0.32), and geomorphology (q = 0.22). This highlights land-use types as the most critical environmental factor determining the spatial pattern of carbon storage. These findings provide scientific insights and recommendations for the sustainable development management and the enhancement of carbon storage functions.

**Data availability statement:** The data underlying this study are available from the Open Science Framework database at https://osf.io/nqyug/(DOI:10.17605/OSF.IO/NQYUG).

**Funding:** Mengting Jin was supported by the Science and Technology Innovation Foundation of the Command Center for Comprehensive Survey of Natural Resources (KC20230015) and the China Geological Survey Project (DD20220962). She played a role in data collection, methodology, and project administration. Quan Xu was supported by the China Geological Survey Project (DD20240740). He played a role in data collection and methodology.

**Competing interests:** The authors have declared that no competing interests exist.

## 1. Introduction

Severe global climate change presents significant challenges to human survival and sustainable development [1,2]. Carbon storage in terrestrial ecosystems constitutes a critical component of global carbon storage, playing a key role in mitigating global climate change by absorbing and releasing greenhouse gases and effectively regulating regional climates [3]. Land-use and cover change (LUCC) is one of the primary driving factors influencing changes in carbon stocks. LUCC impacts vegetation and soil carbon storage in terrestrial ecosystems, thereby alteing regional carbon storage, ecosystem structure, and function, ultimately affecting carbon cycling processes [4,5]. In the context of China's dual carbon strategy—aiming to peak carbon emissions by 2030 and achieve carbon neutrality by 2060—Xinjiang has emerged as a strategically significant region. Therefore, investigating the spatiotemporal patterns of land use and cover changes and their impacts on carbon storage is essential.

Current methods for assessing the impact of LUCC on carbon storage include field surveys and model simulations [6]. Unlike traditional field surveys, model simulations can evaluate carbon stock changes at various scales and provide spatially visualized evaluation results. Among the many available evaluation models (e.g., ARIES, GUMBO, MIMES, CITYgreen), the Integrated Valuation of Ecosystem Services and Tradeoffs (InVEST) model has been widely adopted due to its low data requirements, fast processing speed, and high assessment accuracy [7–9]. Developed by the Natural Capital Project in the United States, InVEST is a modeling system designed to evaluate the functional volume and economic value of ecosystem services, supporting ecosystem management and decision-making. He et al. [10] used a new model to assess the impacts of urban expansion on regional carbon storage by linking the Land Use Scenario Dynamics-urban and InVEST models. Similarly, Wang et al. [11] applied the InVEST model to estimate carbon storage in the Taihang Mountains from 2005 to 2020. They examined the main drivers of the spatial evolution of carbon storage and analyzed their driving mechanisms. These studies highlight that the InVEST model is one of the most widely used approaches for evaluating regional carbon storage.

Exploring the impact of LUCC on ecosystem carbon storage under different scenarios can provide quantified and spatially visualized distribution results. This information is valuable for decision-makers in formulating effective land-use policies and ecological protection plans, as well as optimizing the national spatial pattern [12–14]. However, most existing studies primarily focus on analyzing historical carbon stock changes resulting from land-use alterations, with relatively few studies addressing future land-use changes under multiple scenarios and their impact on carbon storage. Additionally, limited research investigates the spatial heterogeneity of carbon storage using spatial statistics to uncover the influence of diverse driving factors on the spatiotemporal evolution characteristics of carbon storage [15,16].

To address these gaps, we simulated future land-use patterns in Xinjiang under natural growth and ecological protection scenarios using the Future Land Use Simulation (FLUS) model [17], based on land-use data from 2000 to 2020. Subsequently, we employed the InVEST model to evaluate the spatiotemporal evolution of carbon storage in Xinjiang from 2000 to 2020 and predicted conditions for 2035 under different scenarios. Finally, we applied Geodetector to analyze the impact of various driving factors on the spatial distribution of carbon storage.

## 2. Study area and data

### 2.1. Overview of the study area

Xinjiang is located inland, between 73°40′E to 96°18′E longitude and 34°25′N to 48°10′N latitude. The region features complex terrain, including various landforms such as mountains

and basins. It has a typical temperate continental climate characterized by abundant sunshine and significant temperature differences. The area is generally arid with relatively low annual precipitation—averaging approximately 150 mm. Precipitation varies significantly across the region, with northern Xinjiang receiving more rainfall than southern Xinjiang, while temperatures are higher in southern Xinjiang compared to the north [18]. The topography of Xinjiang affects carbon stocks in a number of ways, with vegetation cover and soil conditions in mountainous areas being conducive to carbon fixation and storage, while basins and unutilized areas have relatively low carbon stocks. Factors such as topographic relief, altitude and moisture conditions also influence carbon cycling and distribution to some extent [19].

Xinjiang is China's largest provincial-level administrative region, covering 1.66 million km², which accounts for one-sixth of China's total land area. It has a border length exceeding 5,000 km, sharing boundaries with eight neighboring countries. Historically, Xinjiang was a critical hub of the ancient Silk Road, and today it plays an essential role in China's "Belt and Road" policy. Its strategic geopolitical importance cannot be overstated. Consequently, studying the spatiotemporal evolution of carbon storage in Xinjiang holds significant practical value. A schematic of the study area is provided in Fig 1.

## 2.2. Data

The data used in this study (Table 1) were obtained from the Resource and Environmental Science Data Platform.

The land use data is derived from remote sensing satellite imagery, including Landsat 8 OLI and GF-2, among others. This dataset is acquired through a high-resolution remote sensing technology system that integrates unmanned aerial vehicles and ground surveys, complemented by human-computer interactive interpretation methods grounded in geoscience knowledge. The classification and overall accuracy were assessed using a confusion matrix. The first category of land use achieved a comprehensive accuracy of over 93%, while the second category's comprehensive accuracy exceeded 90%, satisfying the user's requirement

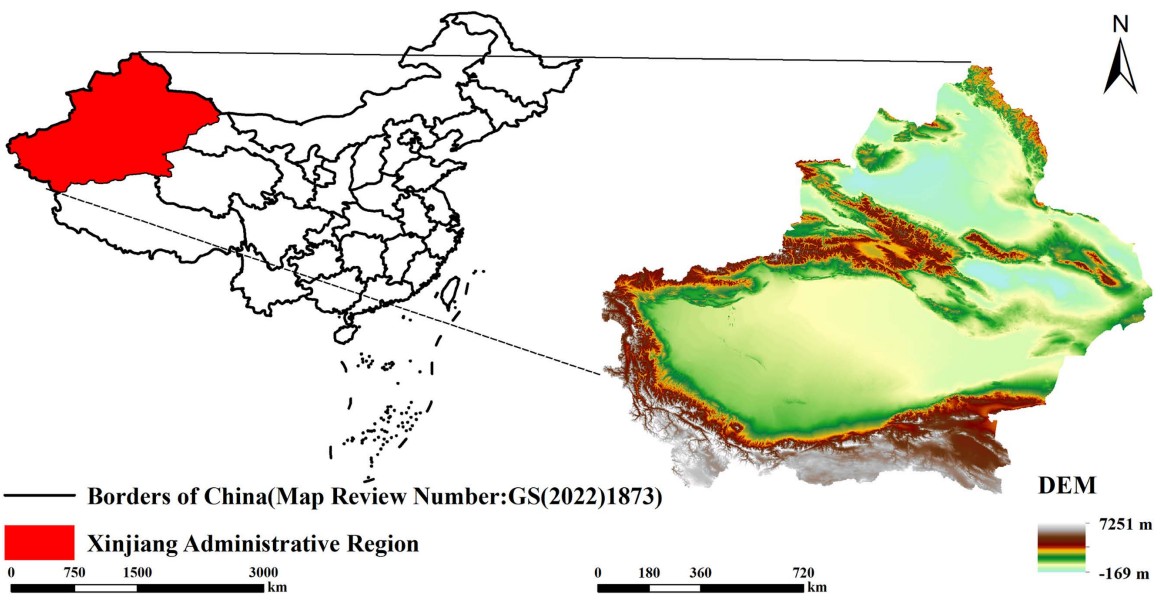

**Fig 1. Summary map of the study area.** The basemap is obtained from the Standard Map Service System (http://bzdt.ch.mnr.gov.cn/).

**Table 1. Data declaration.**

| Data types | Data resolution | Data use | Data sources |
|---|---|---|---|
| Land-use types | 1 km × 1 km | Model base input data | Resource and Environmental Science Data Platform (http://www.resdc.cn/) |
| DEM; Slope; Aspect | 30 m × 30 m | Driving factor | |
| Rain; Temperature; Land surface temperature; Net primary productivity; Luminous index; Soil types; Geomorphology | 1 km × 1 km | Driving factor | |

for a drawing accuracy at a scale of 1:100,000 [20,21]. Based on the "Classification Standard for Land Use Status" (GBT 21010–2017) and accompanying data descriptions, land-use types were categorized into six classes: cropland, forest land, grassland, water area, construction land, and unused land.

Based on previous research [18], the selected driving factors included the Digital Elevation Model (DEM), slope, aspect, rainfall, land surface temperature (LST), net primary productivity (NPP) of vegetation, luminous index, soil types, and geomorphology. To ensure consistency across datasets, the projection method was unified and transformed into the Albers_Conic_Equal_Area system. Additionally, the spatial resolution was resampled to 1 km using cubic convolution interpolation.

## 3. Method

The study involved three key steps: (1) a multi-scenario land use simulation using the FLUS model, (2) calculations of the spatiotemporal characteristics of carbon storage using the InVEST model, and (3) an analysis of the driving forces behind carbon stock using the Geodetector. The methods applied in each of these steps are detailed in the following subsections. The technological roadmap is illustrated in Fig 2.

### 3.1. Multi-scenario land-use change simulation based on the FLUS model

FLUS is a land-use prediction model based on cellular automata [22]. Using transformation rules and quantitative relationships between land-use types and driving factors, the model employs a roulette selection algorithm to simulate the spatial pattern of land use under different scenarios and years [23]. It also spatially allocates simulated land-use demand [24,25]. In this study, we used GeoSOS-FLUS V2.4 software to predict land-use changes in Xinjiang under different scenarios for the year 2035.

Based on existing land-use data, we applied the Markov model to simulate future land-use type areas in Xinjiang for 2035. Driving factors for land-use change included the DEM, slope, aspect, rainfall, temperature, land surface temperature (LST), net primary productivity (NPP) of vegetation, luminous index, soil types, and geomorphology. Notably, temperature refers to the thermal state of the atmosphere, measured in a sheltered environment, which indicates atmospheric warmth. In contrast, LST reflects the ground's thermal condition, measured at the interface between the Earth's surface and the air. LST is influenced by various factors such as terrain type, vegetation, and soil moisture.

We also incorporated neighborhood factors and transfer matrix parameters into the simulation. Neighborhood factors represent the difficulty of converting one land-use type to another, with values ranging from 0 to 1. A score closer to 1 indicates a strong expansion ability for that land-use type. Transfer matrices are represented by values of 0 and 1: a value of

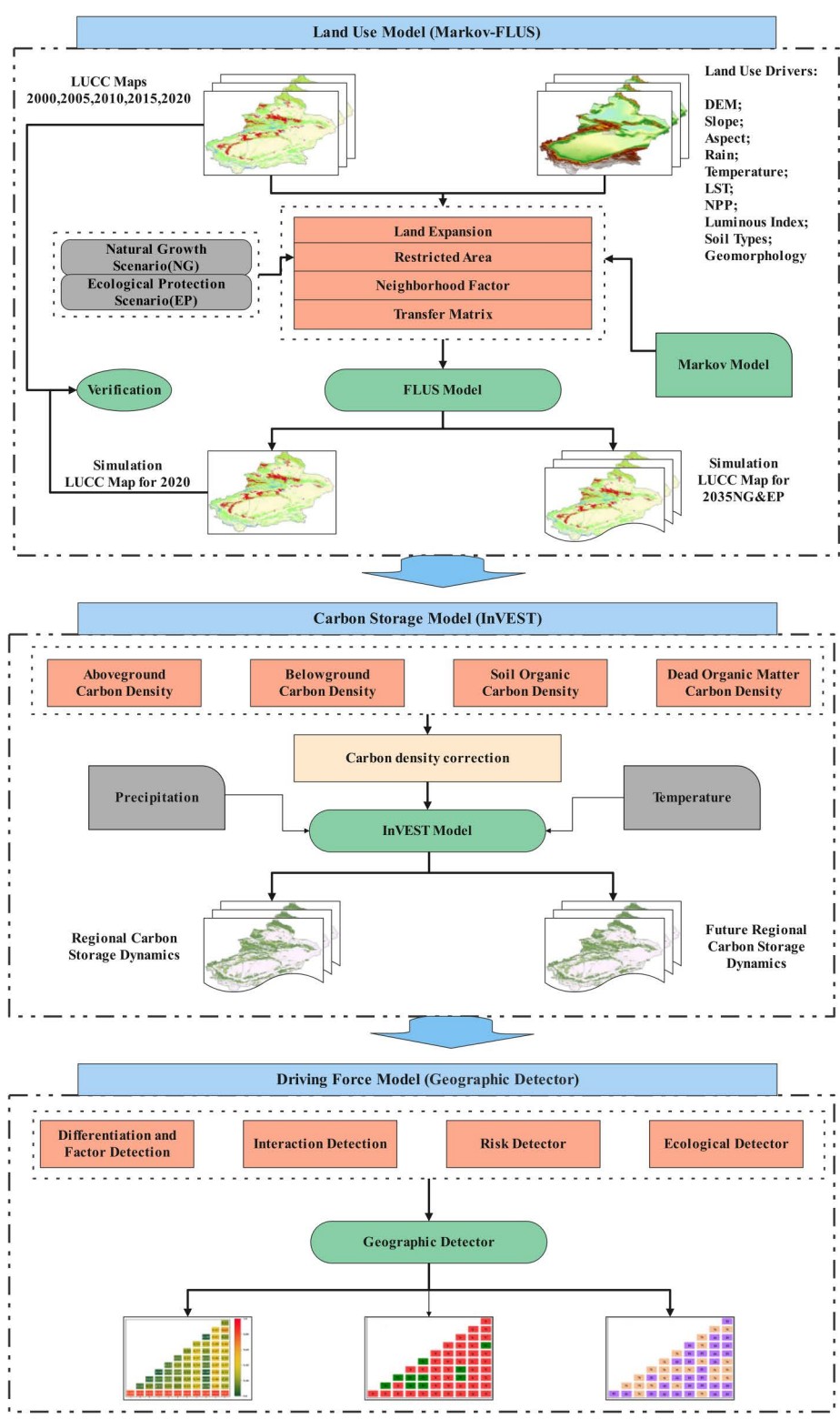

**Fig 2. Technology roadmap.**

1 indicates that a conversion between two land-use types is permissible, while 0 indicates that it is not. Detailed parameter settings are provided in Table 2.

Two land-use scenarios were simulated: natural growth and ecological protection. In the natural growth scenario, land-use types were allowed to evolve based on current conditions without deliberate modification, and the corresponding matrix value was set to 1. In the ecological protection scenario, enhanced protection measures were applied to ecological lands, such as forests, grasslands, and water bodies, while the expansion ability of other land-use types was restricted. For this scenario, the corresponding matrix value was set to 0.

Finally, we estimated the spatial distribution of various land-use types in Xinjiang for 2035 under both scenarios by modifying the input parameters of the FLUS model.

## 3.2. Carbon storage assessment based on the InVEST model

In this study, the carbon storage module of the InVEST model was used to assess the spatio-temporal changes in carbon storage in Xinjiang. This module calculates ecosystem carbon storage by integrating land-use and carbon density data [26]. It categorizes ecosystem carbon storage into four primary carbon pools: aboveground vegetation carbon ($C_{above}$), belowground vegetation carbon ($C_{below}$), soil organic carbon ($C_{soil}$), and dead organic carbon ($C_{dead}$).

Aboveground vegetation carbon pool: Includes the carbon stored in all living vegetation above the ground. Belowground vegetation carbon pool: Refers to the carbon stored in the root systems of living plants. Soil organic carbon pool: Typically represents the organic carbon stored in mineral and organic soils. Dead organic carbon pool: Represents the carbon stored in litter and dead plants [27,28]. The formula for calculating carbon storage is as follows:

$$C = C_{above} + C_{below} + C_{soil} + C_{dead} \qquad (1)$$

$$C_{total} = \sum_{k=1}^{n} A_k \times C_k \qquad (2)$$

**Table 2. Land transfer parameters.**

| Natural growth scenario (NG) | | | | | | |
|---|---|---|---|---|---|---|
| Name | Cropland | Forest land | Grassland | Water bodies | Construction land | Unused land |
| Cropland | 1 | 1 | 1 | 1 | 1 | 1 |
| Forest land | 1 | 1 | 1 | 1 | 1 | 1 |
| Grassland | 1 | 1 | 1 | 1 | 1 | 1 |
| Water bodies | 1 | 1 | 1 | 1 | 1 | 1 |
| Construction land | 1 | 1 | 1 | 1 | 1 | 1 |
| Unused land | 1 | 1 | 1 | 1 | 1 | 1 |
| Weight | 1 | 1 | 1 | 1 | 1 | 1 |
| **Ecological protection scenario (EP)** | | | | | | |
| Name | Cropland | Forest land | Grassland | Water bodies | Construction land | Unused land |
| Cropland | 1 | 1 | 1 | 1 | 0 | 0 |
| Forest land | 1 | 1 | 1 | 1 | 0 | 0 |
| Grassland | 1 | 1 | 1 | 1 | 0 | 0 |
| Water bodies | 1 | 1 | 1 | 1 | 0 | 0 |
| Construction land | 1 | 1 | 1 | 1 | 1 | 1 |
| Unused land | 1 | 1 | 1 | 1 | 1 | 1 |
| Weight | 1 | 1 | 1 | 1 | 0.1 | 0.1 |

Where *C* represents the total carbon storage per unit area for each land-use type, $C_{above}$ represents the carbon density of aboveground biomass, $C_{below}$ represents the carbon density of belowground biomass, $C_{soil}$ represents the soil carbon density, $C_{dead}$ represents the carbon density of dead biomass, and $C_{total}$ represents the total carbon storage of all land cover types. $A_k$ represents the area of each land cover type, $C_k$ represents the carbon density of each unit, and *n* represents the number of land-use types. The carbon density data used in this study were obtained from previous relevant research [29], and the carbon density values for each land-use type are shown in Table 3.

## 3.3. Driving force analysis of spatiotemporal evolution of carbon storage based on Geodetector

The Geodetector is a software tool used to measure and attribute spatially stratified heterogeneity. This method does not make linear assumptions, and it has a simple form with clear physical meaning [30]. The basic idea is to divide the study area into several subregions. If the sum of the variances of the subregions is smaller than the total variance of the region, spatial heterogeneity exists. If the spatial distribution of two variables tends to be consistent, there is a statistical correlation between them [31,32]. The Geodetector includes four detectors:

(1) Differentiation and factor detection

The spatial differentiation of the probe-dependent variable (*Y*) and the extent to which the probe driving factor (*X*) explains the spatial differentiation of *Y*, measured by the q value, is expressed as:

$$q = 1 - \frac{\sum_{h=1}^{L} N_h \sigma_h^2}{N\sigma^2} = 1 - \frac{SSW}{SST} \tag{3}$$

$$S = \sum_{L}^{h=1} N_h \sigma_h^2, \; SST = N\sigma^2 \tag{4}$$

Where: *h* = 1,..., *L* represents the strata of variable *Y* or factor *X*; $N_h$ and *N* are the number of units in layer *h* and the entire region, respectively; $\sigma_h^2$ and $\sigma^2$ are the variances of layer *h* and area *Y* values, respectively; and *SSW* and *SST* are the Within Sum of Squares and Total Sum of Squares, respectively.

(2) Interaction detection

Recognizing the interaction between different driving factors involves assessing the combined effects of two factors (*X1* and *X2*) on the dependent variable *Y* by determining whether

**Table 3. Carbon density data of each land-use type in study area (kg/m².).**

| Land-use types | Aboveground carbon density | Belowground carbon density | Soil organic carbon density |
|---|---|---|---|
| Cropland | 0.1 | 1.45 | 7.95 |
| Forest land | 0.76 | 2.08 | 15.88 |
| Grassland | 0.63 | 1.55 | 8.69 |
| Water bodies | 0.05 | 0 | 0 |
| Construction land | 0.04 | 0 | 0 |
| Unused land | 0.02 | 0 | 2.16 |

their interaction increases or decreases the explanatory power of $Y$, or whether these factors have independent effects on $Y$. The relationships between these two factors can be classified into the following categories:

$$\text{Nonlinearity attenuation: } q(X1 \cap X2) < \text{Min}(q(X1), q(X2))$$

$$\text{Single-factor nonlinearity decreases: } \text{Min}(q(X1), q(X2)) < q(X1 \cap X2) < \text{Max}(q(X1)), q(X2))$$

$$\text{Two-factor enhancement:} q(X1 \cap X2) > \text{Max}(q(X1), q(X2))$$

$$\text{Independent:} q(X1 \cap X2) = q(X1) + q(X2)$$

$$\text{Nonlinear enhancement:} q(X1 \cap X2) > q(X1) + q(X2)$$

(3) Risk detection

A t-test was used to determine whether there was a significant difference between the means of the attributes in two subregions. It was calculated as follows,

$$t_{\overline{y}_{h=1} - \overline{y}_{h=2}} = \frac{\overline{Y}_{h=1} - \overline{Y}_{h=2}}{\left[ \dfrac{Var(\overline{Y}_{h=1})}{n_{h=1}} + \dfrac{Var(\overline{Y}_{h=2})}{n_{h=2}} \right]^{1/2}} \tag{5}$$

Where $\overset{\bullet}{Y}_h$ represents the average value of the attributes in subregion $h$, $n_h$ is the number of samples in subregion $h$, and $V_{ar}$ is the variance.

(4) Ecological detection

The F-test was used to assess whether there was a significant difference in the spatial distribution of attribute Y due to the influence of two factors, X1 and X2. This was achieved by comparing the effects on Y as follows,

$$F = \frac{N_{X1}(N_{x2} - 1)SSW_{X1}}{N_{X2}(N_{x1} - 1)SSW_{X2}} \tag{6}$$

$$SSW_{X1} = \sum\nolimits_{h=1}^{L1} N_h \sigma_h^2 \tag{7}$$

$$SSW_{X2} = \sum\nolimits_{h=1}^{L2} N_h \sigma_h^2 \tag{8}$$

where $N_{X1}$ and $N_{X2}$ represent the sample sizes of the two factors $X1$ and $X2$, respectively; $SSW_{X1}$ and $SSW_{X2}$ denote the sums of the within-group variances formed by $X1$ and $X2$, respectively; and $L1$ and $L2$ represent the numbers of strata for $X1$ and $X2$, respectively.

We used Geodetector Software (Beta) for ArcGlS Pro to investigate the impact of various driving factors on carbon storage in Xinjiang. Carbon storage was the dependent variable ($Y$), and the driving factors were the independent variables ($X$). Random sampling was used to avoid the influence of spatial autocorrelations and human factors. One thousand sample points were randomly generated within the study area, and the carbon storage and driving factor values for each point were extracted for analysis.

## 4. Results

### 4.1. Analysis of characteristics of land use change

Between 2000 and 2020, grassland and unused land were the dominant land-use types in Xinjiang, accounting for more than 28.85% and 60.17% of the total area, respectively (Fig 3 and Table 4). The most common land-use types were cropland, forestland, and water bodies, which accounted for 3.73%, 1.75%, and 2.07% of the total area, respectively. Construction land covered the smallest area, accounting for only 0.27% of the total area. Over the 20 years, cropland, grassland, and construction land showed increasing trends, with increases of 21,712 km², 4,084 km², and 3,371 km², respectively. In contrast, areas of forestland, water bodies, and unused land showed decreasing trends, with decreases of 8,138 km², 11,638 km², and 8,023 km², respectively. From the perspective of spatial distribution, the increases in cropland, grassland, and construction land are mainly concentrated in Tacheng, the Bortala Mongol Autonomous Prefecture, and the oasis areas in southern Xinjiang. The decreases in forest land and water bodies are primarily observed in the Korla and Ili regions.

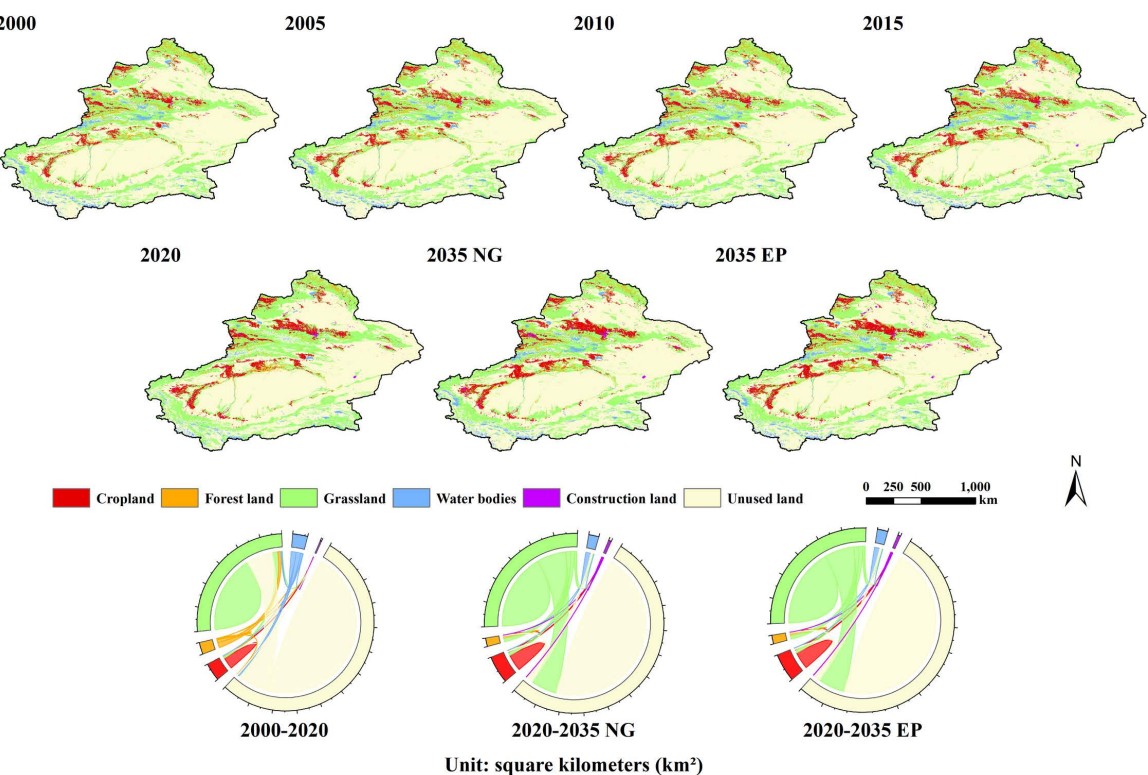

**Fig 3. Map of current land use in Xinjiang and changes in area under different scenarios from 2000 to 2035.**

**Table 4. Land-use type areas in Xinjiang.**

| Land-use types | Area/km² | | | | | | |
|---|---|---|---|---|---|---|---|
| | 2000 | 2005 | 2010 | 2015 | 2020 | 2035NG | 2035EP |
| Cropland | 42611 | 47584 | 63117 | 55251 | 64323 | 69952 | 70085 |
| Forest land | 28111 | 27814 | 35031 | 27148 | 19973 | 25993 | 26390 |
| Grassland | 336894 | 333676 | 424606 | 330195 | 340978 | 321006 | 321186 |
| Water bodies | 35350 | 35463 | 45002 | 35549 | 23712 | 35562 | 35549 |
| Construction land | 3132 | 3431 | 4470 | 4868 | 6503 | 7771 | 7125 |
| Unused land | 696063 | 694193 | 884353 | 691442 | 688040 | 681687 | 681636 |

Concerning land-use transfer, the increase in cropland and forestland was mainly due to the development of unused land, while the increase in construction land resulted primarily from the conversion of cropland, grassland, and unused land. Most forestland was converted to grassland, and areas of water bodies gradually decreased and became unused land. It is evident that, with the continuous development of the social economy between 2000 and 2020, Xinjiang focused on expanding farmland, conducting afforestation, and constructing infrastructure. Areas of unused land gradually decreased, accelerating urbanization and playing a positive role in Xinjiang's economic development and environmental protection. However, due to the impacts of human activities and climate change, there have also been some negative effects on ecological lands such as forestland and water bodies.

In the 2035 natural growth scenario, cropland, forestland, water bodies, and construction land in Xinjiang are expected to increase by 5,629 km², 6,020 km², 11,850 km², and 1,268 km², respectively, with increases of 8.75%, 30.14%, 49.97%, and 19.50%. Areas of grassland and unused land are expected to decrease by 19,972 km² and 6,353 km², respectively, with decreases of 5.86% and 0.92%. From the perspective of land-use transfer, the increase in cropland, forestland, and construction land will mainly occur through the conversion of grassland. In contrast, the increased water body areas will mainly result from the conversion of grassland and unused land. Grassland areas are predicted to decrease significantly, primarily due to their conversion into unused land.

Under the ecological protection scenario, the change in the land-use pattern in Xinjiang was predicted to be similar to that under the natural growth scenario, but with different area changes. Compared to the natural growth scenario, cropland, forestland, and grassland would increase by 133 km², 397 km², and 180 km², respectively, while water bodies, construction land, and unused land would decrease by 13 km², 646 km², and 51 km², respectively. Under the ecological protection scenario, croplands, forestlands, and grasslands, which are the primary contributors to carbon storage, were effectively protected to a some extent.

## 4.2. Characteristics analysis of carbon storage changes under multiple scenarios

The spatial distribution of carbon storage in Xinjiang is shown in Fig 4. In terms of quantity, the total amount of carbon stored in Xinjiang every 5 years from 2000 to 2020 was $8689.373 \times 10^5$ t, $8692.023 \times 10^5$ t, $8697.929 \times 10^5$ t, $8700.356 \times 10^5$ t, and $8826.889 \times 10^5$ t, respectively, exhibiting an overall increasing trend. There was a cumulative increase of $137.515 \times 10^5$ t, with a growth rate of 1.58%. On average, the amount of carbon stored increased by $6.876 \times 10^5$ t per year. The most significant change in Xinjiang's carbon storage occurred between 2015 and 2020, with an increase of $126.532 \times 10^5$ t, accounting for 92.01% of the cumulative increase. In terms of area, the area of carbon stock increase amounted to 188,928

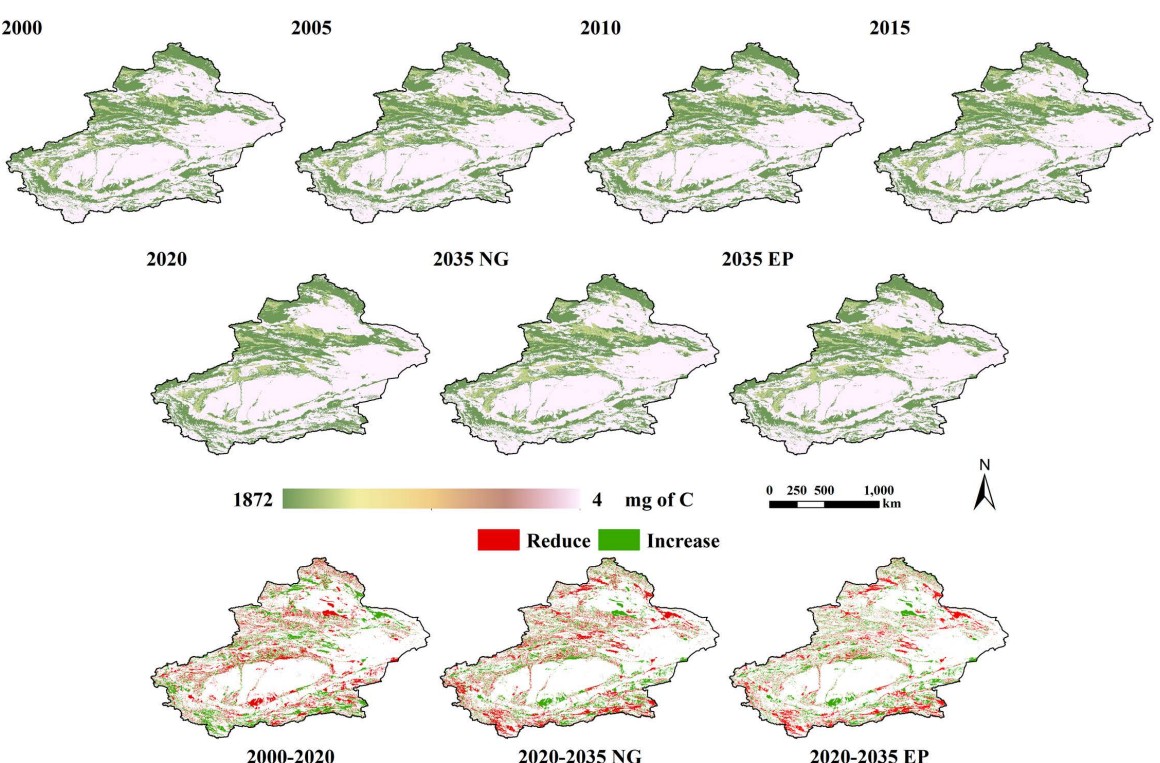

**Fig 4. Status of carbon storage in Xinjiang from 2000 to 2035 and changes under different scenarios.**

km² and the area of decrease amounted to 182,667 km². From a spatial distribution perspective, regions demonstrating significant increases in carbon storage are predominantly concentrated in the mountainous areas of Xinjiang. In contrast, areas showing decreased carbon storage levels are primarily distributed along the periphery of both the Taklimakan Desert and the Gurbantunggut Desert in Xinjiang.

As shown in Fig 5, high carbon storage areas were mainly concentrated in cropland, forestland, and grassland areas due to the different land-use types with varying carbon sequestration capabilities. In contrast, the carbon densities of water bodies, construction land, and unused land were relatively low, resulting in a lower carbon storage content. Between 2000 and 2020, the increased cropland and grassland areas in Xinjiang resulted in a relative increase in carbon storage. The changes in carbon storage were mainly influenced by changes in land-use types.

Under the natural growth scenario, the overall trend of carbon storage in Xinjiang is expected to decrease, with a projected reduction of $168.344 \times 10^5$ t in 2035 compared to 2020. This decline in carbon storage is primarily attributed to the conversion of large areas of grassland into construction land and unused land, leading in a decrease in both soil carbon storage and the carbon storage of aboveground and belowground vegetation. As a result, the total amount of carbon stored in Xinjiang would decline. However, the ecological protection scenario would mitigate this reduction in carbon storage. By limiting the conversion of forestland and grassland to other land-use types, carbon storage could increase by $13.227 \times 10^5$ t compared to the natural growth scenario. The implementation of ecological protection policies could enhance the effectiveness of regional ecological conservation and facilitate sufficirnt carbon sequestration in Xinjiang.

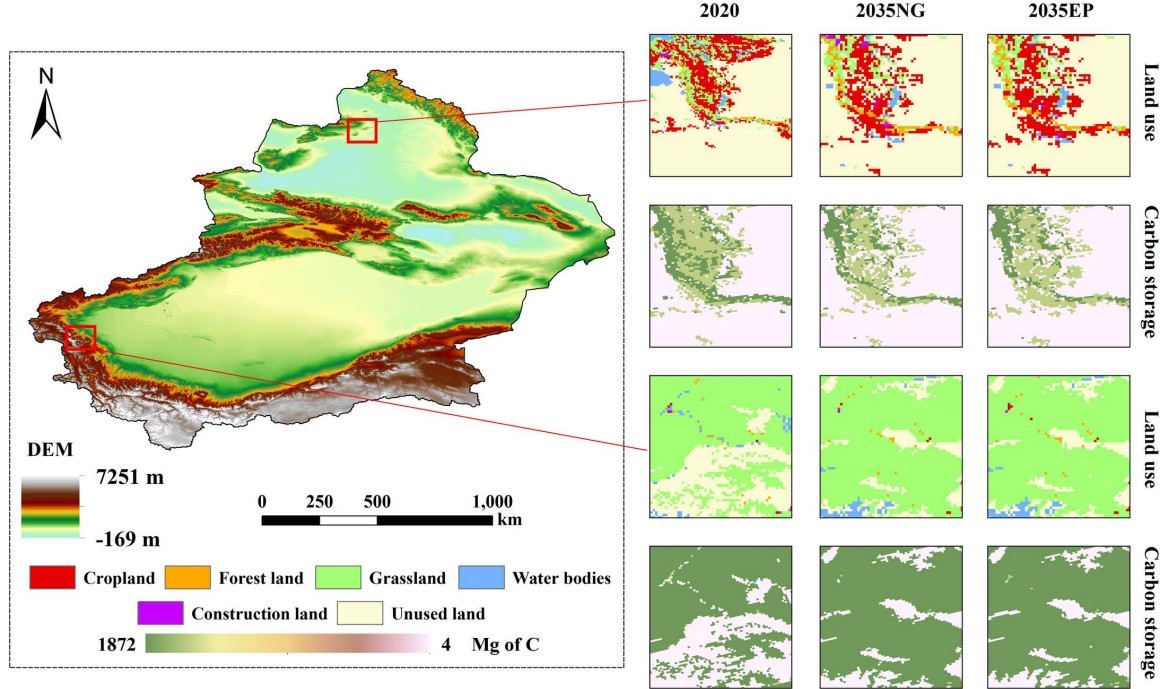

**Fig 5. Local details of carbon storage.**

## 4.3. Driving force analysis of carbon storage based on Geodetector

The results of the analysis based on the Geodetector are shown in Fig 6. Fig 6a presents the calculated results for all driver divergence probes. The divergence factor q ranges from 0 to 1, with larger values indicating stronger explanatory power of the driver regarding carbon stock, and vice versa. The results revealed that land-use types had the highest q value (0.80), followed by soil types (0.41), NPP (0.32), and geomorphology (0.22), while the q values for the other factors were relatively small. Furthermore, from the perspective of interaction detection, the q-value increased to 0.83 after the interactions between land-use types and both soil types and geomorphological factors. This further enhanced the explanatory power of carbon storage, suggesting that these factors jointly play a positive role.

Fig 6b presents the interaction types among all driving factors, enabling an assessment of whether the combined effect of any two factors enhances or reduces the explanatory power for carbon stocks. The results indicate that pairwise interactions between all factors can produce an enhancing effect. Specifically, land-use types serve as a two-factor enhancer when interacting with DEM, slope, rain, temperature, LST, NPP, soil types, and geomorphology, respectively. In contrast, land-use types act as a nonlinear enhancer when interacting with aspect and luminous index, respectively. All these interactions contribute to a further improvement in the explanatory power of carbon stocks.

Fig 6c presents the ecological detection results for all the driving factors, which indicate whether there was a significant difference in their impact on the spatial distribution of carbon storage. ("*Y*" indicates a significant difference, while "*N*" indicates no significant difference.) From the results, it can be seen that land-use types, LST, NPP, and soil types are all significantly different from other factors, indicating that these types of factors act independently on carbon stock distribution, respectively.

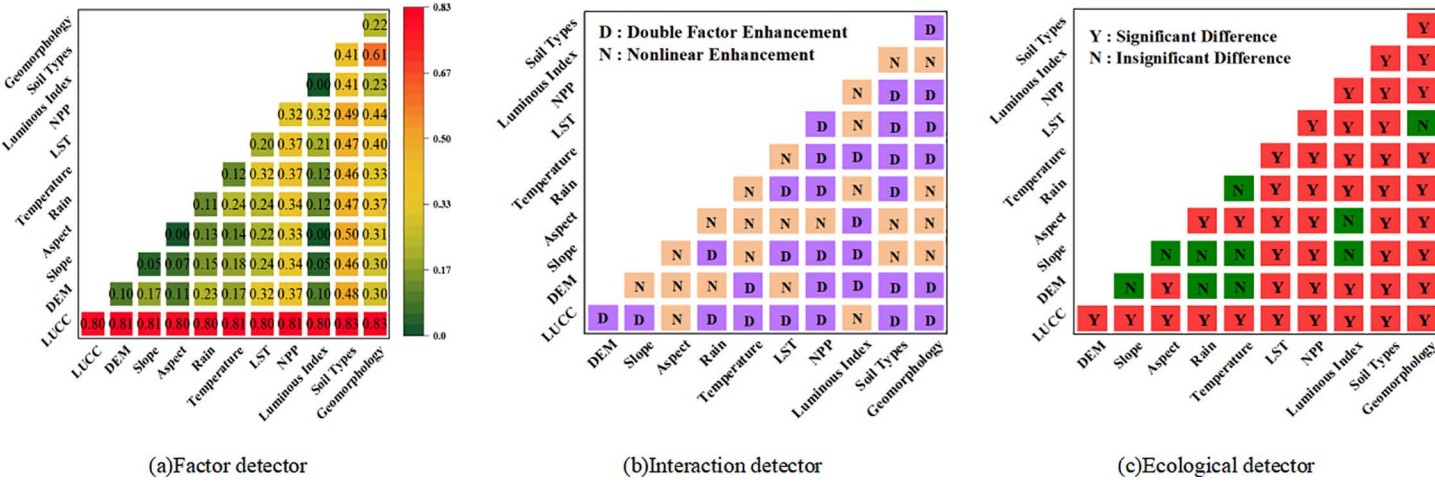

**Fig 6. Map displaying the outcomes of the Geodetector's analysis on the drivers of carbon stocks in Xinjiang.**

In conclusion, the analysis showed that, among these driving factors, land-use type was the most relevant and robust explanatory factor for change in carbon storage, and it was the primary environmental factor determining the spatial pattern of carbon storage.

## 5. Discussion

Rapid urbanization leads to large-scale encroachment of construction land on croplands, forests, and grasslands [33]. From 2000 to 2020, the overall trend of carbon storage in Xinjiang increased. However, under the two typical future scenarios, the carbon storage trend is expected to decrease. Employing the ecological protection scenario would mitigate this trend to some extent compared to the natural growth scenario because it comprehensively coordinates the functions of various land types and is more conducive to slowing the loss of carbon storage. Implementing such protectionist policies would help maintain a regional carbon balance and be of great significance for optimizing the spatial pattern of Xinjiang's land.

The results showed that between 2000 and 2020, grassland was the dominant land-use type in Xinjiang, and its carbon storage capacity was higher than that of other ecosystems. Therefore, protecting grasslands is key to achieving the sustainable development of carbon storage functions in Xinjiang. The changes in carbon storage under different scenarios show that Xinjiang should prioritize protecting its grassland resources to prevent degradation. It is essential to focus on optimizing the land-use structure with a low-carbon orientation and to promote the transformation of low-coverage grassland and unused land into high-coverage grassland through policies that establish grazing bans and protection zones, restrict associated land development, and strengthen the legal management of grasslands. Additionally, it is crucial to improve the grassland monitoring network and consolidate the achievements of grassland ecological restoration.

Land-use change is a complex process influenced by multiple driving factors [34,35]. In this study, factors such as DEM, slope, aspect, precipitation, and temperature were selected as the driving factors for simulating future land-use patterns. These factors exhibited an excellent fitting effect on the various land-use types, and the simulated results were highly accurate (Kappa = 0.94), thus meeting the research requirements. Our results were supported by those of previous relevant studies[29].

                                                     

In future research, policy-related factors should be added into the driving factor system to reveal the impact of land use policies on carbon storage. At the same time, more driving factors should be further screened and optimized to improve the accuracy of land-use simulation and driving force analysis. Moreover, to accurately estimate regional carbon storage changes and enhance the accuracy of model verification, efforts should be made to strengthen the acquisition of carbon density data, as well as to conduct localization calibration and field measurements of core indicators.

## 6. Conclusions

(1) From 2000 to 2020, the land-use types in Xinjiang were dominated by grassland and unused land, which accounted for more than 28.85% and 60.17% of the total area, respectively. By 2035, cropland, forestland, water bodies, and construction land are expected to increase, while areas of grassland and unused land are expected to decrease. Under the ecological protection scenario, cropland, forestland, and grassland—key contributors to carbon storage—will receive adequate protection. Compared to the natural growth scenario, the areas of these land types are expected to increase by 133 km$^2$, 397 km$^2$, and 180 km$^2$, respectively.

(2) From 2000 to 2020, the overall trend of carbon storage in Xinjiang increased, with a cumulative rise of $137.515 \times 10^5$ t, or an increase of 1.58%, and an average annual increase of $6.876 \times 10^5$ t. By 2035, carbon storage is expected to decrease, with an estimated reduction of $168.344 \times 10^5$ t compared to 2020. Ecological protection would help mitigate this decrease in carbon storage: under the ecological protection scenario, carbon storage is expected to increase by $13.227 \times 10^5$ t compared to the natural growth scenario.

(3) A Geodetector was used to explore the degree of influence each driving force on the spatial distribution of carbon storage. The results showed that land-use type had the highest explanatory power for carbon storage (q = 0.80), followed by soil types (0.41), NPP (0.32), and geomorphology (0.22). These findings indicate that land-use type is the primary environmental factor determining the spatial pattern of carbon storage.

## Author contributions

**Conceptualization:** Mengting Jin, Quan Xu.

**Data curation:** Mengting Jin, Quan Xu.

**Formal analysis:** Mengting Jin, Quan Xu.

**Investigation:** Yunfei Zhang.

**Methodology:** Mengting Jin, Quan Xu.

**Project administration:** Mengting Jin, Quan Xu.

**Resources:** Mengting Jin, Quan Xu.

**Supervision:** Yunfei Zhang.

**Validation:** Yunfei Zhang.

**Writing – original draft:** Mengting Jin.

**Writing – review & editing:** Xingxing Duan, Yunfei Zhang, Quan Xu.

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
