## [Decision Letter · Decision Letter 0]

10 Dec 2024

PONE-D-24-34190Predicting the Spatial Pattern of Land Use Change and Carbon Storage in Xinjiang: A Markov-FLUS-InVEST Model ApproachPLOS ONE

Dear Dr. Xu,

Thank you for submitting your manuscript to PLOS ONE. After careful consideration, we feel that it has merit but does not fully meet PLOS ONE’s publication criteria as it currently stands. Therefore, we invite you to submit a revised version of the manuscript that addresses the points raised during the review process.

We look forward to receiving your revised manuscript.

Kind regards,

Sudeshna Bhattacharjya, Ph.D

Academic Editor

PLOS ONE

**Journal Requirements:**

Mengting Jin was supported by the Science and Technology Innovation Foundation of the Command Center for Comprehensive Survey of Natural Resources (KC20230015) and the China Geological Survey Project (DD20220962). She played a role in data collection, methodology, and project administration. Quan Xu was supported by the China Geological Survey Project (DD20240740). He played a role in data collection and methodology. 

3. We note that Figures 1, 2, 3, 4 and 5 in your submission contain map images which may be copyrighted. All PLOS content is published under the Creative Commons Attribution License (CC BY 4.0), which means that the manuscript, images, and Supporting Information files will be freely available online, and any third party is permitted to access, download, copy, distribute, and use these materials in any way, even commercially, with proper attribution. For these reasons, we cannot publish previously copyrighted maps or satellite images created using proprietary data, such as Google software (Google Maps, Street View, and Earth). For more information, see our copyright guidelines: http://journals.plos.org/plosone/s/licenses-and-copyright.

We require you to either present written permission from the copyright holder to publish these figures specifically under the CC BY 4.0 license, or remove the figures from your submission:

a. You may seek permission from the original copyright holder of Figures 1, 2, 3, 4 and 5 to publish the content specifically under the CC BY 4.0 license.  

Reviewers' comments:

Reviewer's Responses to Questions

**Comments to the Author**

1. Is the manuscript technically sound, and do the data support the conclusions?

Reviewer #1: Partly

Reviewer #2: Yes

2. Has the statistical analysis been performed appropriately and rigorously? 

Reviewer #1: No

Reviewer #2: Yes

3. Have the authors made all data underlying the findings in their manuscript fully available?

Reviewer #1: No

Reviewer #2: Yes

4. Is the manuscript presented in an intelligible fashion and written in standard English?

Reviewer #1: Yes

Reviewer #2: Yes

5. Review Comments to the Author

**Reviewer #1: ** How the LULC maps for 2000, 2005, 2010, 2015, 2020 were generated? Using which classification method and satellite data? The accuracies during calibration and validation phase with confusion matrix should be presented.

What are the input parameters which were modified to generate future land use? What are the sources of those future input parameters? Details required.

What about the neighborhood factors for Natural growth scenario and ecological protection scenario?

Which software were used for future land use and driving force analysis? Please mention that clearly with library or package name.

What do you mean by “geographic detector”? Is it a model? If yes, what is the name of the model? What are the inputs and output of the model?

Abstract

Expand “FLUS-InVEST”. Do not use abbreviations in abstract.

L No. 28: Driving factors include which parameters? Please specify.

Introduction

L No. 77-84: Delete “However, most existing studies have…factors on the spatiotemporal evolution characteristics of carbon storage.” As it is a repetition of L No. 69-76.

L No. 90-93: Delete “The results of this study provide scientific… and ecological decision-making.”

Figure 1: Replace ‘KM’ with ‘km’ in the scalebar.

Table 1 Write the source of DEM, rain, temperature, LST, NPP, nighttime light index, soil type, landform type with URL link.

L No. 125: Which interpolation method was used?

Replace “water areas” to “water bodies” throughout the manuscript.

Fig. 2 What do you mean by agrotype? Please maintain uniformity of the factor names throughout the manuscript. In table 1 - nighttime light index is written as Luminous index and landform type is written as geomorphology.

Table 2, 3, 4; Figure 3 and 5: “Arable land” to “Cropland”, “Woodland” to “Forest land”

Please expand the figure caption of figure 3 and 4 to make it self-explanatory.

L No. 279: What do you mean by “high-value carbon storage areas”?

Fig. 6: Please enlarge the figures to make it legible.

**Reviewer #2: ** The structure of this paper is clear and the methods used are flexible. It begins by analyzing the changes in land use in Xinjiang from 2000 to 2020, then simulates and predicts the land use in Xinjiang in 2035 under two scenarios using the FLUS-InVEST Model, and also analyzes the spatial and temporal changes in carbon storage in Xinjiang. Finally, it uses geographic model detectors to analyze the impact of driving factors. But it is recommended that the content be revised prior to publication so as to improve the quality of the article and its accessibility to readers. The main issues are as follows:

1. Are there any incorrect statements in the summary of results? Based on the research findings, the driving factors had the greatest carbon storage explanatory power (q=0.80), "The geographical detector analysis results showed that driving factors had the greatest carbon storage explanatory power (q=0.80)".

2. In the Introduction, can you provide some models in the "among the many evaluation models" section to support your argument?

3. The sections from Introduction 69-76 and 77-84 are repetitive in meaning. Please modify this part and delete the redundant sentences. Also, please appropriately cite relevant literature to support your viewpoint.

4. The scale of the maps in Figure 1. Summary map of the study area and Figure 5. Local details of carbon storage should be "km". Please change it.

5. The DEM legend in Figure 1. Summary map of the study area and Figure 5. Local details of carbon storage does not have units. Is it necessary to specify units for greater precision?

6. Please explain the principle behind the 0 and 1 in Table 2 for different scenario settings.

7. In section 3.1, please explain the difference between "temperature" and "land surface temperature (LST)" in the selected driving factors.

8. How were the driving factors chosen when using the Markov model in this study, and how were the driving factors chosen when building the geographical model detector?

9. Reference 20 has an incorrect citation format.

10. Why is there only a string diagram for land use transfer in Figure 3 for the natural growth scenario, but not for the ecological protection scenario?

11. In "Xinjiang plays a vital role in global carbon storage because of its complex geological structure and mountain chains that include the Altai Mountains, Kunlun Mountains, and Tianshan Mountains", is it a bit stiff and disconnected? Suggest enriching the theoretical framework before introducing the topic.

6. PLOS authors have the option to publish the peer review history of their article (what does this mean? ). If published, this will include your full peer review and any attached files.

**Do you want your identity to be public for this peer review?** For information about this choice, including consent withdrawal, please see our Privacy Policy .

Reviewer #1: **Yes: ** Bappa Das

Reviewer #2: No

---

## [Author Response · Author response to Decision Letter 1]

21 Jan 2025

I would like to express my heartfelt respect and gratitude to the expert for your meticulous and professional revision in your busy schedule, which makes the article more rigorous and further improved. Thanks to the editor and experts for the opportunity to revise. If there is any problem, please feel free to contact me at any time. I am very willing to make positive changes.

Reviewer #1

1. How the LULC maps for 2000, 2005, 2010, 2015, 2020 were generated? Using which classification method and satellite data? The accuracies during calibration and validation phase with confusion matrix should be presented.

Response 1: LULC maps come from the Resource and Environmental Science Data Platform. This data is based on remote sensing satellite images, Landsat 8 OLI, GF-2 and other remote sensing satellite data, and the land use data set is obtained through the construction of high-resolution remote sensing - unmanned aerial vehicle - ground survey observation technology system, combined with the human-computer interactive interpretation method based on geoscience knowledge. The classification accuracy and total accuracy were evaluated by confusion matrix. The comprehensive evaluation accuracy of the first type of land use is more than 93%, and the comprehensive accuracy of the second type classification is more than 90%, which meets the user's drawing accuracy of 1: 100,000 scale. We have combined the opinions of expert to provide a detailed explanation of the data sources in the article.

[1] Kuang W, Zhang S, Du G, et al. Monitoring Periodically National Land Use Changes and Analyzing Their Spatiotemporal Patterns in China During 2015-2020. Journal of Geographical Sciences. 2022; 32(9), 1705-1723.

[2] Liu J, Kuang W, Zhang Z, et al. Spatiotemporal Characteristics, Patterns, and Caus-es of Land-use Changes in China Since the Late 1980s Journal of Geographical Sciences. 2014; 24(2): 195-210.

2. What are the input parameters which were modified to generate future land use? What are the sources of those future input parameters? Details required.

Response 2: Input parameters include DEM, slope, aspect, rainfall, land surface temperature, net primary productivity of vegetation, luminous index, soil type, and geomorphology. These Data are from the Resource and Environmental Science Data Platform. We have made further supplementary explanations on the data sources in the manuscript in combination with the opinions of expert. Please see "2.2 Data" and "3.1 Multi-scenario land-use change simulation based on the FLUS model" for details.

3. What about the neighborhood factors for Natural growth scenario and ecological protection scenario?

Response 3: Neighborhood factors represented the difficulty of converting a land-use type into other land-use types, with a parameter range of 0-1. A score close to 1 indicates that the land-use type has a strong expansion ability. The natural growth scenario does not involve any intervention, so the neighborhood factor is set to 1. The ecological protection scenario is to strengthen the protection of forest land, grassland, water and other ecological land, while weakening the expansion capacity of other land types. Therefore, set the other land type parameter to 0.1. At the same time, we have made further additions to the presentation based on expert comments. Please see section 3.1 for details.

4. Which software were used for future land use and driving force analysis? Please mention that clearly with library or package name.

Response 4: We used GeoSOS-FLUS V2.4 Software for future land use simulation and Geodetector Software (beta) for ArcGlS Pro software for driving force analysis. We have taken the expert's advice in full and supplemented the package names in this article.

5. What do you mean by “geographic detector”? Is it a model? If yes, what is the name of the model? What are the inputs and output of the model?

Response 5: Geographic detector is a model called Geodetector. The model's input comprises the carbon stock value in Xinjiang, serving as the dependent variable (Y), alongside the values of ten distinct independent variable drivers (X1-X10). The output yields a contribution index for each driver, thereby quantifying the respective impact of these variables on the carbon stock. We have supplemented Geodetector with expert recommendations. For detailed modifications, refer to Section 3.3: Driving force analysis of spatiotemporal evolution of carbon storage based on Geodetector.

6. Abstract：

Expand “FLUS-InVEST”. Do not use abbreviations in abstract.

L No. 28: Driving factors include which parameters? Please specify.

Response 6: We have fully adopted the expert opinion and changed the abbreviation to the full name in the abstract. Furthermore, the driving factors were changed to specific parameters (land-use types). Please see the abstract section for details.

7. Introduction：

L No. 77-84: Delete “However, most existing studies have…factors on the spatiotemporal evolution characteristics of carbon storage.” As it is a repetition of L No. 69-76.

L No. 90-93: Delete “The results of this study provide scientific… and ecological decision-making.”

Response 7: We fully adopted the opinions of experts and deleted this part of the content.

8.Figure 1: Replace ‘KM’ with ‘km’ in the scalebar.

Response 8: We have replaced “KM” in the scale bar with “km” as recommended by expert.

9.Table 1 Write the source of DEM, rain, temperature, LST, NPP, nighttime light index, soil type, landform type with URL link.

Response 9: We have fully adopted the expert opinion and added source links for all the above data in Table 1.

10.L No. 125: Which interpolation method was used?

Response 10: Our resampling method used cubic convolution interpolation. We have provided additional explanations in the text. Detailed modification details can be found in "2.2. Data".

11.Replace “water areas” to “water bodies” throughout the manuscript.

Response 11: We fully adopted the expert opinion and replaced “water areas ” with “water bodies” throughout the manuscript.

12.Fig. 2 What do you mean by agrotype? Please maintain uniformity of the factor names throughout the manuscript. In table 1 - nighttime light index is written as Luminous index and landform type is written as geomorphology. Table 2, 3, 4; Figure 3 and 5: “Arable land” to “Cropland”, “Woodland” to “Forest land”

Response 12: The agrotype in the figure represents the soil types. We have redrawn the diagram and modified it to soil types. In addition, we have uniformly modified the full-text factor names according to expert recommendations.

13.Please expand the figure caption of figure 3 and 4 to make it self-explanatory.

Response 13: We have fully adopted the expert's advice and changed the title of the figure to make it more detailed and self-explanatory.

14.L No. 279: What do you mean by “high-value carbon storage areas”?

Response 14: “High-value carbon storage areas” means areas with high carbon storage. For better expression, we changed it to “high carbon storage areas”.

15.Fig. 6: Please enlarge the figures to make it legible.

Response 15: We have adopted expert advice and have enlarged the text in the figure.

Reviewer #2

The structure of this paper is clear and the methods used are flexible. It begins by analyzing the changes in land use in Xinjiang from 2000 to 2020, then simulates and predicts the land use in Xinjiang in 2035 under two scenarios using the FLUS-InVEST Model, and also analyzes the spatial and temporal changes in carbon storage in Xinjiang. Finally, it uses geographic model detectors to analyze the impact of driving factors. But it is recommended that the content be revised prior to publication so as to improve the quality of the article and its accessibility to readers. The main issues are as follows:

1.Are there any incorrect statements in the summary of results? Based on the research findings, the driving factors had the greatest carbon storage explanatory power (q=0.80), "The geographical detector analysis results showed that driving factors had the greatest carbon storage explanatory power (q=0.80)".

Response 1: Thanks for the expert advice. We have revised the statements in the summary of the results. “The geographical detector analysis results showed that land-use types had the greatest carbon storage explanatory power (q=0.80).”

2. In the Introduction, can you provide some models in the "among the many evaluation models" section to support your argument?

Response 2: We have fully followed the expert's advice and supplemented the introduction. Among the many models, including ARIES, GUMBO, MIMES, CITYgreen [1].

[1] Ma L, Jin T, Wen Y, Wu X, Liu G. The Research Progress of InVEST Model. Ecological Economy. 2015; 31(10), 126-131+179.

3. The sections from Introduction 69-76 and 77-84 are repetitive in meaning. Please modify this part and delete the redundant sentences. Also, please appropriately cite relevant literature to support your viewpoint.

Response 3: We have fully adopted the expert's advice, deleted the repetitive sentences in the introduction, and quoted some literature to support our viewpoint.

[1] Liang Y, Hu H, Crowther TW, Jörgensen RG, Liang C, Chen J, et al. Global decline in microbial-derived carbon stocks with climate warming and its future projections. National Science Review. 2024; 11: nwae330.

[2] Zhu K, He J, Tian X, Hou P, Wu L, Guan D, et al. Analysis of Evolving Carbon Stock Trends and Influencing Factors in Chongqing under Future Scenarios. Land. 2024; 13: 421.

4. The scale of the maps in Figure 1. Summary map of the study area and Figure 5. Local details of carbon storage should be "km". Please change it.

Response 4: Based on expert opinions, we have changed "KM" in all figures to "km".

5. The DEM legend in Figure 1. Summary map of the study area and Figure 5. Local details of carbon storage does not have units. Is it necessary to specify units for greater precision?

Response 5: We have fully adopted the expert opinions and redrawn the figures as required.

6. Please explain the principle behind the 0 and 1 in Table 2 for different scenario settings.

Response 6: We have fully adopted the expert's advice and made further supplementary explanations on the principle behind it. The transfer matrix is represented by 0 and 1. When conversion of one land use type to another is allowed, the corresponding value of the matrix is set to 1 and 0 when it is not allowed. The natural growth scenario does not involve any intervention, so the value is set to 1. The ecological protection scenario aims to strengthen the protection of ecological land such as forests, grasslands, and water bodies, so it does not allow them to change to other land use types, so it is set to 0. Please see section 3.1 for details.

7. In section 3.1, please explain the difference between "temperature" and "land surface temperature (LST)" in the selected driving factors.

Response 7: We have fully adopted the expert's advice. The difference between "temperature" and "Surface temperature (LST)" was added to section 3.1.

Temperature denotes the air's thermal state, specifically the air temperature recorded in a sheltered environment, which indicates the level of atmospheric warmth.

Land surface temperature pertains to the ground's thermal condition. It is measured at the boundary between the Earth's surface and the air and is influenced by various factors, including the type of terrain, vegetation, and soil moisture.

8.How were the driving factors chosen when using the Markov model in this study, and how were the driving factors chosen when building the geographical model detector?

Response 8: We are grateful to the expert for invaluable comments, which have charted the course for our future research endeavors. Following the expert' recommendations, we have implemented additional revisions within the domains of data, methodology, and discussion. The choice of driving factors when utilizing the Markov model was primarily informed by established research. Through rigorous experimentation, our study achieved a high degree of simulation precision (Kappa = 0.94), satisfying the stringent research criteria. Moreover, with the aim of enhancing the model's predictive accuracy and offering a robust framework for future factor selection, we employed the geographical detector to meticulously quantify the individual impact of the above driving factors on carbon stock levels. Given the intricate nature of carbon stock assessment, our future investigations will encompass a broader spectrum of considerations, including legislative and policy-oriented aspects.

9.Reference 20 has an incorrect citation format.

Response 9: We have modified the format of the 20th reference based on expert advice and have checked other references.

10.Why is there only a string diagram for land use transfer in Figure 3 for the natural growth scenario, but not for the ecological protection scenario?

Response 10: According to the opinions of expert, we have redrawn Figure 3 and added the string diagram of land use transfer under the ecological protection scenario. At the same time, we also modified Figure 4 to add the carbon storage change chart under the ecological protection scenario. Please see Figures 3 and 4 for detailed changes.

11.In "Xinjiang plays a vital role in global carbon storage because of its complex geological structure and mountain chains that include the Altai Mountains, Kunlun Mountains, and Tianshan Mountains", is it a bit stiff and disconnected? Suggest enriching the theoretical framework before introducing the topic.

Response 11: According to the suggestions of expert, we have deleted this part of the content and modified and improved the introduction of the study area. In addition, we also describe the theoretical framework in more detail before introducing the topic. Details can be found in 2.1 Overview of the study area.

Thanks again for the expert's contribution to this manuscript.

---

## [Decision Letter · Decision Letter 1]

18 Feb 2025

PONE-D-24-34190R1Predicting the spatial pattern of land use change and carbon storage in Xinjiang: A Markov-FLUS-InVEST model approachPLOS ONE

Dear Dr. Xu,

Thank you for submitting your manuscript to PLOS ONE. You can see that reviewers have positive comments regarding your manuscript. However, I would like to request you to consider the comment of second reviewer to incorporate minor modifications and submit a revised version of the manuscript that addresses the points raised during the review process.

We look forward to receiving your revised manuscript.

Kind regards,

Sudeshna Bhattacharjya, Ph.D

Academic Editor

PLOS ONE

Journal Requirements:

Reviewers' comments:

Reviewer's Responses to Questions

**Comments to the Author**

1. If the authors have adequately addressed your comments raised in a previous round of review and you feel that this manuscript is now acceptable for publication, you may indicate that here to bypass the “Comments to the Author” section, enter your conflict of interest statement in the “Confidential to Editor” section, and submit your "Accept" recommendation.

Reviewer #1: All comments have been addressed

Reviewer #2: All comments have been addressed

2. Is the manuscript technically sound, and do the data support the conclusions?

Reviewer #1: Yes

Reviewer #2: Yes

3. Has the statistical analysis been performed appropriately and rigorously? 

Reviewer #1: Yes

Reviewer #2: Yes

4. Have the authors made all data underlying the findings in their manuscript fully available?

Reviewer #1: Yes

Reviewer #2: Yes

5. Is the manuscript presented in an intelligible fashion and written in standard English?

Reviewer #1: Yes

Reviewer #2: Yes

6. Review Comments to the Author

Reviewer #1: The authors have responded to all my questions satisfactorily. The revised version may be accepted for publication.

Reviewer #2: The author has earnestly considered and meticulously revised the manuscript in response to the reviewers' comments. After examination, I am fundamentally satisfied with the revisions made by the author in the article. Through the revisions, the quality of the article has also enhanced. Nevertheless, it is proposed that the author could appropriately depict the characteristics of spatial distribution and incorporate regional variations in the analysis, rather than merely focusing on the description of land types.

Finally, it is recommended to accept the article for publication, but the author is requested to carry out the final polishing based on the above remaining issues. I believe that after these adjustments, the article will be more consummate.

7. PLOS authors have the option to publish the peer review history of their article (what does this mean? ). If published, this will include your full peer review and any attached files.

**Do you want your identity to be public for this peer review?** For information about this choice, including consent withdrawal, please see our Privacy Policy .

Reviewer #1: **Yes: ** Bappa Das

Reviewer #2: No

---

## [Author Response · Author response to Decision Letter 2]

21 Feb 2025

I would like to express my heartfelt respect and gratitude to the expert for your meticulous and professional revision in your busy schedule, which makes the article more rigorous and further improved. Thanks to the editor and experts for the opportunity to revise. If there is any problem, please feel free to contact me at any time. I am very willing to make positive changes.

Journal Requirements:

Response 1: We would like to express our sincere appreciation to the editor for providing us with the opportunity to revise our manuscript. We have conducted a thorough review of all references and implemented the necessary revisions in compliance with the specified guidelines. Upon further investigation, it was noted that references 9 and 10 are not indexed in the Web of Science database; however, they are available through the CNKI (China National Knowledge Infrastructure). As a result, we have replaced these references with suitable alternatives, ensuring that the original text's integrity and content remain intact.

Reviewer #1

1. The authors have responded to all my questions satisfactorily. The revised version may be accepted for publication.

Response 1: We are sincerely grateful to the esteemed reviewer for granting us this invaluable opportunity, which we consider both a privilege and a profound responsibility.

Reviewer #2

1.The author has earnestly considered and meticulously revised the manuscript in response to the reviewers' comments. After examination, I am fundamentally satisfied with the revisions made by the author in the article. Through the revisions, the quality of the article has also enhanced. Nevertheless, it is proposed that the author could appropriately depict the characteristics of spatial distribution and incorporate regional variations in the analysis, rather than merely focusing on the description of land types. Finally, it is recommended to accept the article for publication, but the author is requested to carry out the final polishing based on the above remaining issues. I believe that after these adjustments, the article will be more consummate.

Response 1: In response to the reviewers' valuable comments, we have substantially revised Sections 4.1 and 4.2 by incorporating detailed descriptions of the spatial distribution patterns and regional heterogeneity of both land use and carbon storage. We are deeply grateful for the reviewers' insightful suggestions, which have considerably improved the scientific rigor and overall quality of our manuscript.

Thanks again for the expert's contribution to this manuscript.

---

## [Decision Letter · Decision Letter 2]

14 Mar 2025

Predicting the spatial pattern of land use change and carbon storage in Xinjiang: A Markov-FLUS-InVEST model approach

PONE-D-24-34190R2

Dear Dr. Xu,

We’re pleased to inform you that your manuscript has been judged scientifically suitable for publication and will be formally accepted for publication once it meets all outstanding technical requirements.

Kind regards,

Sudeshna Bhattacharjya, Ph.D

Academic Editor

PLOS ONE

Additional Editor Comments (optional):

Reviewers' comments:

Reviewer's Responses to Questions

**Comments to the Author**

1. If the authors have adequately addressed your comments raised in a previous round of review and you feel that this manuscript is now acceptable for publication, you may indicate that here to bypass the “Comments to the Author” section, enter your conflict of interest statement in the “Confidential to Editor” section, and submit your "Accept" recommendation.

Reviewer #2: All comments have been addressed

2. Is the manuscript technically sound, and do the data support the conclusions?

Reviewer #2: Yes

3. Has the statistical analysis been performed appropriately and rigorously? 

Reviewer #2: Yes

4. Have the authors made all data underlying the findings in their manuscript fully available?

Reviewer #2: Yes

5. Is the manuscript presented in an intelligible fashion and written in standard English?

Reviewer #2: Yes

6. Review Comments to the Author

Reviewer #2: I have carefully reviewed the revised manuscript submitted to the Journal. I would like to express my appreciation to the authors for their thorough and detailed responses to the review comments.

The authors have addressed all the comments I raised. Each point has been carefully considered, and corresponding modifications have been made to the manuscript.

Overall, the revisions have greatly improved the quality of the manuscript. I recommend that it be accepted for publication.

7. PLOS authors have the option to publish the peer review history of their article (what does this mean? ). If published, this will include your full peer review and any attached files.

**Do you want your identity to be public for this peer review?** For information about this choice, including consent withdrawal, please see our Privacy Policy .

Reviewer #2: No

---

## [Editor Report · Acceptance letter]

PONE-D-24-34190R2

PLOS ONE

Dear Dr. Xu,

I'm pleased to inform you that your manuscript has been deemed suitable for publication in PLOS ONE. Congratulations! Your manuscript is now being handed over to our production team.

Kind regards,

on behalf of

Dr. Sudeshna Bhattacharjya

Academic Editor

PLOS ONE